# Centroid-Aware Gaussian Prompt Learning with Erosion-Guided Accumulator for Robust Semantic Cell Segmentation

**Nur Suriza Syazwany**                                    SURIZASYAZWANY@INHA.EDU

**Su Jung Kim**                                            SK2266@INHA.EDU

**Ju-Hyeon Nam**                                           JHNAM0514@INHA.EDU

**Sang-Chul Lee**                                          SCLEE@INHA.AC.KR

*Department of Electrical and Computer Engineering*
*Inha University, South Korea*

## Abstract

Accurately segmenting cell images remains challenging due to variations in cell size, shape, and overlapping structures. Existing approaches often struggle with densely packed and overlapping cell regions, leading to inconsistent performance. While recent methods, such as the Segment Anything Model (SAM), have shown promise, they rely heavily on manual prompting, which can be time-consuming and inconsistent for densely packed nuclei datasets. To address these limitations, we propose a novel centroid-guided Gaussian map-based accumulator prompting approach for robust nuclei segmentation. Our method constructs Gaussian maps by accumulating centroids across multiple erosion iterations, capturing the frequency and spatial distribution of nuclei centroids. These maps serve as informative priors to guide the segmentation model, enhancing its ability to localize cell structures while maintaining adaptability to varying cell sizes. By integrating these Gaussian-based prompts into a transformer-based segmentation model, our approach enables refined predictions with improved spatial awareness. We validate our method on two challenging, densely packed datasets, DSB18 and ConSeP, demonstrating robust and superior segmentation performance over state-of-the-art methods.

**Keywords:** cell segmentation, prompt

## 1 Introduction

Precise segmentation is essential for applications such as cancer diagnosis, drug discovery and the examination of cellular behaviors in health and disease (Zhang et al. (2025); Chen et al. (2024)). However, robust segmentation remains challenging due to inherent variability in cell shapes, sizes, imaging artifacts (Mouelhi et al. (2018)), and are often clustered, further complicating the task of distinguishing individual cells (Schmidt et al. (2018)).

Recent advances in deep learning have significantly transformed cell nuclei segmentation, with tailored network architectures and supervision strategies. U-Net (Ronneberger et al. (2015)) set a benchmark in medical image segmentation with its skip connections properties. However, UNet and its variants (Long (2020); Li et al. (2022); Nam et al. (2024)) still struggle with overlapping nuclei and delineating boundaries in noisy images (Avazov et al. (2024)). HoverNet(Graham et al. (2019)) and Cpp-net(Chen et al. (2023)) addressed these

---

. Code is available on `https://github.com/ejawany95/accumulator_gausian_prompt_cellseg`

issues using distance and boundary maps, but rely on complex post-processing that adds computational overhead. More recently, transformer-based methods have shown promise in medical imaging, following the success of Vision Transformer (Dosovitskiy et al. (2020)). Hybrid models like TransUNet (He et al. (2023)) and CellVIT (Hörst et al. (2024)) integrate self-attention with CNNs to improve spatial reasoning. Meanwhile, Pyramid Vision Transformer v2 (PVTv2) (Wang et al. (2022)) stands out for its efficient multi-scale representation, making it well-suited for the segmentation task (Dong et al. (2021)), though its use in cell segmentation has not yet been explored.

Most histopathology models (Zhou et al. (2019); Lal et al. (2021)) rely solely on image data, unlike human experts who incorporate contextual reasoning, spatial relationships and auxiliary cues such as centroids and boundaries. This gap highlights an opportunity to enhance segmentation methods by integrating auxiliary information that mimics the human cognitive system. The Segment Anything Model (SAM) (Kirillov et al. (2023)) demonstrates the power of prompt-based generalization, yet SAM relies heavily on interactive prompting, limiting its practicality in dense nuclei scenarios. Prompt learning (Liu et al. (2023)) enhances the adaptability of pretrained large models with task-specific prompts, showing effectiveness across visual tasks (Khattak et al. (2023); Wang et al. (2023)). However, existing methods often depend on user-defined inputs such as points or boxes (Kirillov et al. (2023); Ma et al. (2024)), which can be ambiguous in crowded settings. SPPNet (Xu et al. (2023)) use single-point centroids, but still struggle with overlapping cells. Meanwhile, methods that leverage anchor points (Shui et al. (2024)) or density maps (Lou et al. (2024)) remain computationally intensive and require ground truth annotations.

To overcome current limitations, we propose a centroid-guided approach using Gaussian map-based accumulator prompts for nuclei segmentation. Unlike methods relying on raw images or manual prompts, our model automatically generates probabilistic Gaussian maps from centroid data accumulated over erosion iterations. These maps highlight likely cell centers, serving as spatial priors that improve the model's ability to segment densely clustered or overlapping nuclei. The Gaussian prompts are integrated into a UNet-like architecture with PVTv2 encoders, guiding the model's attention toward regions of high nuclei probability. To highlight the robustness of our method under challenging conditions, we focus on two benchmark datasets, ConSeP and Data Science Bowl 2018 (DSB18), both known for their high cellular density and presence of miscellaneous cells (Tyagi et al. (2023); Lee et al. (2022)). These datasets provide a realistic benchmark for assessing the robustness of our method in highly crowded and visually complex environments. The contribution of this paper can be summarized as follows:

- We propose an accumulator-based Gaussian map generation method that automatically identifies centroid locations, significantly reducing human bias and eliminating the need for manual prompting.

- We employ a cross-attention mechanism to fuse these Gaussian prompts directly with the encoder's multi-scale features. This targeted fusion enables the model to dynamically emphasize regions with high nuclei probabilities, enhancing segmentation accuracy, particularly in challenging scenarios with densely packed or overlapping nuclei.

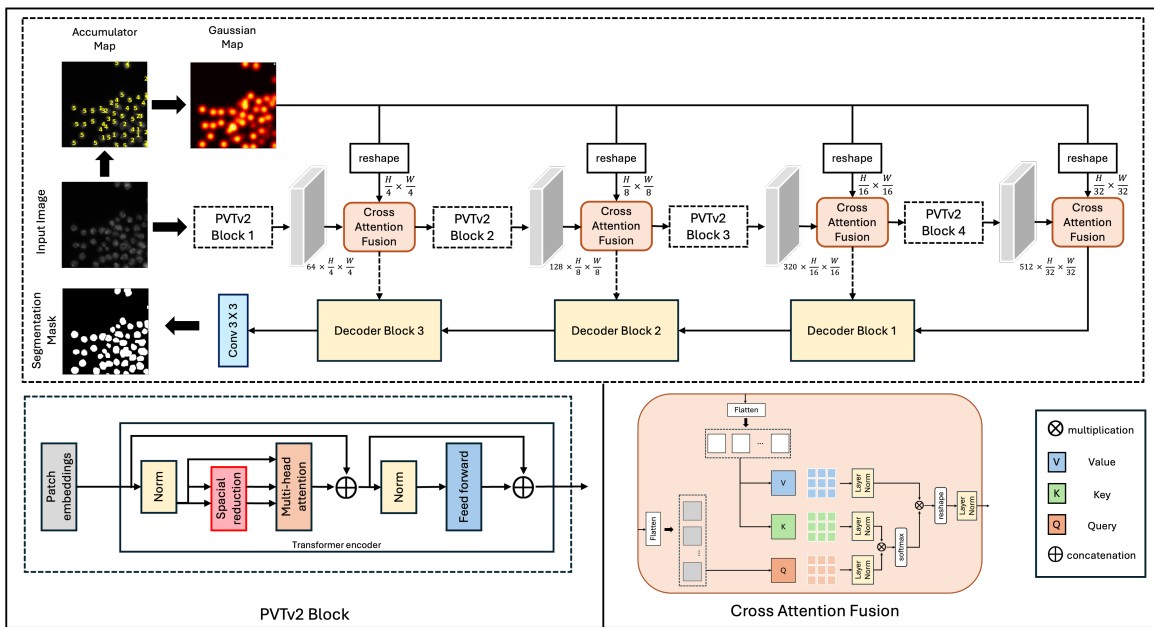

Figure 1: Overview of the proposed architecture. Gaussian maps are generated as spatial priors from input images and fused with PVTv2 encoder features via cross-attention, where priors serve as keys and values and encoder features as queries. The fused features are propagated through the encoder-decoder via skip connections to produce the final segmentation map.

- We evaluate our model on two densely populated cell datasets, demonstrating robust segmentation performance across diverse and challenging imaging conditions.

## 2 Proposed Method

In this work, we introduce Centroid-Aware Gaussian Prompt Learning with Erosion-Guided Accumulator, a novel approach that leverages Gaussian Maps as an auxiliary modality to enhance nucleus localization. As shown in Fig. 1, our model follows a UNet-like encoder-decoder architecture with two input streams: an image stream and a Gaussian prompt stream. A hierarchical feature extraction backbone, combined with a cross-attention fusion module, integrates image and Gaussian map features at multiple scales. The enhanced feature representations are then processed by a multi-scale upsampling decoder to generate the final segmentation mask.

### 2.1 Accumulator-based Gaussian Map Generation

Accurately identifying cell structures poses significant challenges due to overlapping cells and heterogeneous morphological features. To address these challenges, we propose an Accumulator-Gaussian map approach, as illustrated in Fig. 2. This method leverages an iterative centroid accumulator to record centroid frequency across eroded images, refining it with a Gaussian function to produce a continuous, probabilistic cell distribution. This Gaussian map acts as a powerful spatial prior, guiding the model's attention toward likely nuclei regions and significantly enhancing its ability to distinguish cellular structures.

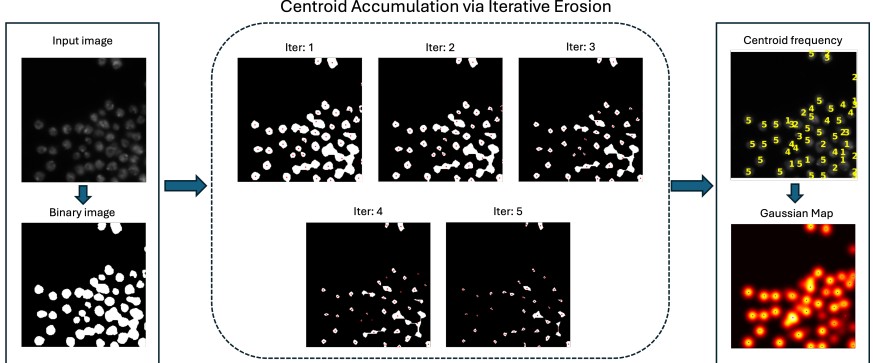

Figure 2: Centroid accumulation using iterative erosion. A binary image is eroded iteratively, and centroids are tracked to build an accumulator map. A Gaussian function is applied to convert this into a probabilistic map, where higher intensities indicate frequent centroid locations.

**Image Preprocessing**: The image is first converted to grayscale and smoothed using Gaussian blurring. Next, Otsu's thresholding is applied to binarize the image, which is generated directly from the input image rather than the ground truth mask, ensuring its applicability in real clinical scenarios.

**Accumulator Construction via Iterative Erosion**: Generating centroids directly from binarized cell images is challenging due to noise and overlapping cells. To overcome this, we implement a method that employs iterative erosions along with an accumulator. With each iteration, the binary mask, $I_e$, is progressively refined, separating clustered regions, thereby facilitating more accurate centroid detection:

$$\mathcal{I}_e^{(i+1)} = \mathcal{I}_e^{(i)} \ominus K \tag{1}$$

Here, $\ominus$ denotes erosion, $K$ is a 3×3 kernel and $i$ indicates each iteration in the erosion process.

After each iteration step, we extract the centroids from the connected components of the refined mask. The set of centroids is defined as follows:

$$C_i = \{(x_j, y_j)\}_{j=1}^{N_i} \tag{2}$$

where $N_i$ represents the number of cells detected in the $i$-th iteration, and $(x_j, y_j)$ represents the horizontal and vertical coordinates of the $j^{th}$ centroid. These centroids are mapped onto an accumulator grid of size $\frac{H}{s} \times \frac{W}{s}$, where $s$ is the grid cell size, set to 8. For each centroid $(x_j, y_j) \in C_i$, its corresponding grid coordinates are computed and used to update the accumulator:

$$A^i(g_x, g_y) = A^{i-1}(g_x, g_y) + 1, \quad g_x = \left\lfloor \frac{x_j}{s} \right\rfloor, \quad g_y = \left\lfloor \frac{y_j}{s} \right\rfloor \tag{3}$$

$A^i(g_x, g_y)$ represents the frequency of centroid occurrences within grid cell $(g_x, g_y)$, after $i^{th}$ iterations. The process is repeated for a total of T=5 iterations, resulting in final accumulator value $A^T(g_x, g_y) \in [0, T]$. The accumulator quantifies centroid likelihood by measuring the consistency of their occurrences across multiple erosions, producing a structured density map of probable cell centroids.

To match the original image size, each accumulator cell is expanded into a pixel block of corresponding area, forming an aligned accumulator map, $A'$. This preserves spatial relationships while restoring the full image size from the low-resolution grid.

**Gaussian Map Generation**: Following accumulator construction, a Gaussian function is applied at each non-zero location $(g_x, g_y)$ in the aligned accumulator, generating a Gaussian heatmap:

$$G(x,y) = \sum_{(g_x, g_y) \in \mathcal{C}} A'(g_x, g_y) \cdot \exp\left(-\frac{(x - g_x)^2 + (y - g_y)^2}{2\sigma^2}\right), \tag{4}$$

where $C = \{(g_x, g_y) \mid A'(g_x, g_y) > 0\}$ is the set of locations with nonzero centroid density, and $\sigma$ controls the spread of the Gaussian distribution. This transformation smooths the centroid influence across the image, creating a continuous spatial representation, where high-intensity regions indicate frequently occurring centroids. By incorporating this probabilistic representation, our proposed method not only identifies the centroids but also provides explicit spatial cues that guide the segmentation network towards densely clustered nuclei.

## 2.2 Encoder and Decoder

Our model leverages PVTv2-B2 as the encoder backbone, which extract hierarchical feature maps at four different scales $\mathcal{F}_i \in \mathbb{R}^{\frac{H}{2^{i+1}} \times \frac{W}{2^{i+1}} \times Ch_i}$, where $Ch_i \in \{64, 128, 320, 512\}$ and $i \in \{1, 2, 3, 4\}$, given an input image $I \in \mathbb{R}^{H \times W \times 3}$.

To inject spatial priors into segmentation, we use a Gaussian Map G, derived from the centroid accumulator, which encodes the likelihood of cell centroids and highlights dense regions. For compatibility in feature fusion, G is resized to match the resolution of each feature map by interpolation. The resulting attention-enhanced features are projected back to their original dimensions using an additional $1 \times 1$ convolution, normalized using Layer Normalization, $LN$, and then added residually to the original feature map.

The decoder gradually upsamples the encoded features and merges them with corresponding features from the encoder to refine spatial details. This fusion is followed by convolutional layers that enhance feature representations at each stage. At the final step, a convolutional layer followed by a sigmoid activation is applied to produce the binary segmentation map. The model is trained using Binary Cross-Entropy loss.

## 3 Experiment

### 3.1 Datasets

To evaluate the effectiveness of our proposed method, we conduct experiments on two benchmarks: Data Science Bowl 2018 (Caicedo et al. (2019)) and CoNSeP (Graham et al. (2019)). These datasets are known for their considerable challenges, including densely packed and overlapping nuclei, significant variation in cell morphology, and complex background textures. Such characteristics make them ideal benchmarks for evaluating robustness in real-world cell segmentation tasks. For DSB2018 dataset, we use 428 training, 108 validation and 138 test images. The CoNSeP dataset is divided into 445 training, 87 validation and 124 test images.

### 3.2 Implementation Details

Our experiments are conducted using an NVIDIA RTX 3090 Ti GPU with PyTorch 1.11. The model is trained for 200 epochs using the Adam optimizer with a learning rate of 0.0001

| | Method | Params(M) | FLOPs(G) | DSB18 | | Consep | | Average | |
|---|---|---|---|---|---|---|---|---|---|
| | | | | mIoU | DSC | mIoU | DSC | mIoU | DSC |
| Non SAM-based | UNet | 8.64 | 490.62 | 82.44 (0.9) | 89.16 (0.7) | 64.18 (1.3) | 77.24 (1.2) | 73.31 | 83.20 |
| | HoverNet | 46.97 | 8293.44 | 84.03 (0.5) | 90.66 (0.3) | 68.77 (0.7) | 80.90 (0.6) | 76.40 | 85.78 |
| | TransUNet | 105.32 | 60.03 | 76.46 (1.4) | 86.02 (0.9) | 62.11 (1.2) | 75.82 (1.0) | 69.29 | 80.92 |
| | HDA-Net | 139.70 | 3779.76 | 83.61 (0.6) | 89.88 (0.6) | 67.54 (1.1) | 79.83 (1.0) | 75.58 | 84.86 |
| | CellViT | 46.7 | 4127.52 | 80.70 (1.5) | 88.29 (1.0) | 56.02 (3.6) | 70.71 (3.5) | 68.36 | 79.50 |
| SAM-based | SAM | 90.69 | 41.85 | 76.64 (1.5) | 85.93 (1.0) | 63.17 (1.0) | 76.98 (0.8) | 69.91 | 81.46 |
| | SPP-Net | 10.13 | 74.58 | 66.94 (2.9) | 76.53 (2.4) | 51.90 (0.7) | 66.79 (0.7) | 59.42 | 71.66 |
| | PromptNucSeg | 90.69 | 41.85 | 76.67 (1.9) | 85.79 (1.6) | 62.85 (0.3) | 76.77 (0.2) | 69.76 | 81.28 |
| | MedSAM | 4.06 | 41.85 | 62.17 (0.9) | 75.37 (0.7) | 39.63 (1.9) | 55.38 (1.9) | 50.90 | 65.37 |
| | **OURS (with accumulator)** | 31.08 | 44.63 | 84.24 (0.2) | 90.86 (0.1) | 68.87 (0.2) | 81.07 (0.1) | 76.56 | 85.97 |
| | OURS (w/o accumulator) | 31.08 | 44.63 | 84.12 (0.3) | 90.79 (0.2) | 68.65 (0.3) | 80.88 (0.2) | 76.39 | 85.84 |
| | OURS (GT) | 31.08 | 44.63 | 84.36 (0.1) | 90.94 (0.1) | 68.87 (0.2) | 81.06 (0.1) | 76.62 | 86.00 |

Table 1: Quantitative results on two datasets: DSB18 and ConSep. Red indicates best performance, and Blue indicates the second best.

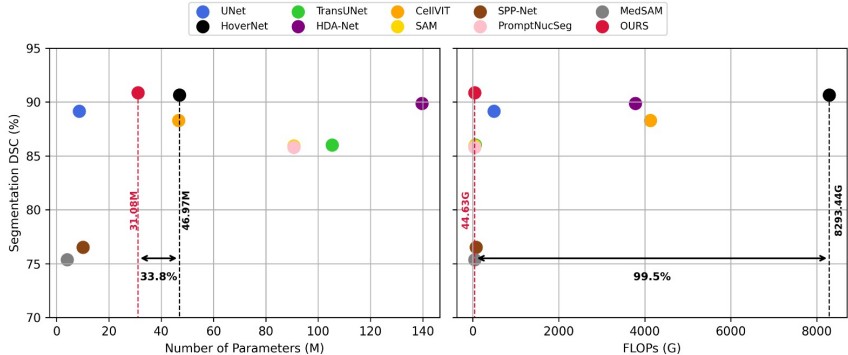

Figure 3: Comparison of efficiency vs performance (DSC) on DSB18.

and a batch size of 4. All input images are resized to $256 \times 256$ to maintain consistency across different datasets.

### 3.3 Comparison with State-of-the-art models

We compared our proposed method with five representative cell segmentation models, including UNet, HoverNet, TransUNet, HDA-Net (Im et al. (2024)), and CellVIT, and four SAM-based models, including SAM, SPP-Net (Xu et al. (2023)), PromptNucSeg (Shui et al. (2024)), and MedSAM (Ma et al. (2024)).

As shown in Tab 1, our method achieves robust performance under both datasets, demostrating its effectiveness across challenging conditions. As shown in Fig. 3, our method requires only 31.08M parameters and 44.63G FLOPs, demonstrating clear advantages in computational efficiency. When compared with HoverNet, the representative model for cell segmentation, our method demonstrates notable gains on both datasets with improvement of 0.16% and 0.19% on average with much fewer FLOPs. Furthermore, our method also surpasses SAM while requiring three times fewer parameters despite not relying on manual prompts or ground truth masks.

To demonstrate that our input image-derived prompts offer performance comparable to ground truth-derived guidance, we also evaluate our method using GT-derived centroids as

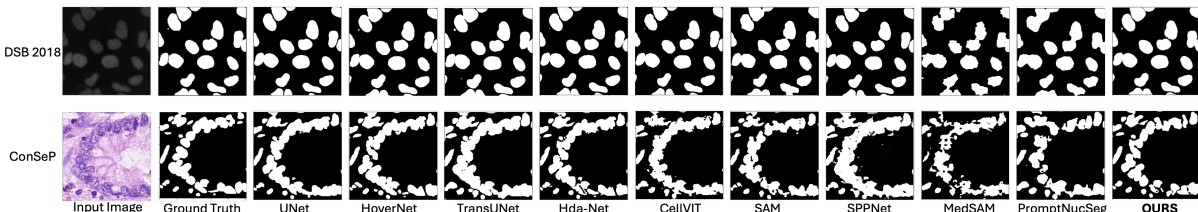

Figure 4: Qualitative result comparison between our model performance against the state-of-the-art methods on DBS18 and ConSep.

shown in Tab. 1. Result shows that GT-based version achieves only marginal improvement (DSB18) or similar (ConSeP), highlighting the reliability of our approach even without relying on manual prompts or ground truth masks.

Qualitative results in Fig. 4 further demonstrate the effectiveness of our method in handling noisy and challenging inputs. In the case of the ConSeP dataset, SAM-based methods like SPPNet and PromptNucSeg struggle to separate nuclei in regions with densely-packed and overlapping centroids. In contrast, our accumulator-driven approach consistently yields clearer boundaries and more precise segmentation.

We also evaluated robustness to common image degradations (Gaussian Blur, Gaussian Noise, JPEG compression) in DSB18, simulating artifacts in microscopy and medical imaging as in (Nam et al. (2025)), as shown in Tab.2. While SAM shows stronger performance under Gaussian Noise, due to its reliance on ground truth masks for prompt generation, our method does not rely on manual or GT-based prompts and achieves more stable performance across Gaussian Blur and JPEG compression, highlighting its robustness to input degradation in a more realistic setting.

| Model | Gaussian Blur | Gaussian Noise | JPEG Compression |
|---|---|---|---|
| UNet | 88.60 | 22.08 | 84.59 |
| SAM | 85.85 | **47.58** | 83.18 |
| Ours | **90.20** | 29.76 | **85.84** |

Table 2: DSC (%) under different robustness conditions. **Bold** is the best performance.

### 3.4 Ablation Studies

We conducted ablation studies to evaluate the impact of Gaussian map parameters and integration strategies on segmentation performance, focusing on three aspects: erosion iteration and $\sigma$ value (Fig. 5) and usage of accumulator in the Gaussian map generator (Tab. 1).

**Varying Sigma Values for Gaussian Map** To investigate the impact of Gaussian kernel spread on centroid-based spatial priors, we evaluated multiple $\sigma$ values, e.g., 3, 5, 7, and 9, each controlling the extent of spatial influence around cell centroids. Smaller values (e.g., $\sigma$=3 or 5) produce narrower priors that are more focused but may increase the model's sensitivity to noise or centroid misalignment. On the other hand, larger values (e.g., $\sigma$=9) result in overly diffuse priors, potentially causing excessive blurring and reduce boundary precision. We found that $\sigma$=7 strikes the best balance, providing spatially informative guidance without overwhelming the segmentation with unnecessary spread.

**Effect of Erosion Iterations** We conducted an ablation study to determine the optimal number of erosion iterations, $T$, for robust centroid detection. Iteration values of 1,

3, 5 and 7 were evaluated to assess their influence on the quality of the generated spatial priors. Fewer iterations (e.g., 1 or 3) did not sufficiently separate overlapping or clustered nuclei, resulting in inaccurate centroid estimates and degraded spatial guidance. In contrast, higher iterations (e.g., 7) led to excessive erosion, potentially removing smaller nuclei or shrinking regions too aggressively, thereby reducing the reliability of the centroid map. Setting the iteration to 5 provided an effective balance, resolving the dense clusters while preserving the cell structure, leading to more accurate centroid localization.

**Usage of Accumulator in Gaussian Map Generator** To assess the effectiveness of the accumulator in our Gaussian map generation process, we conducted an ablation study comparing model variants with and without the accumulator module. In the variant without the accumulator, Gaussian maps were generated directly from the initial binary masks before erosion. This led to less accurate centroid localization, particularly in regions with overlapping nuclei. In contrast, using the accumulator allowed the model to progressively integrate spatial cues, resulting in improved IoU and Dice scores. This highlights the accumulator's role in enhancing segmentation performance through more reliable spatial guidance.

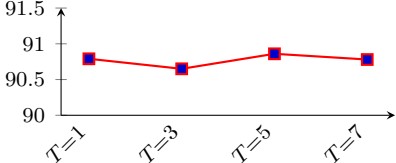 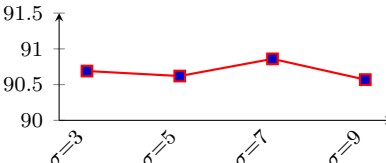

Figure 5: Comparison of model performance (Dice Scores) across training iterations (left) and different $\sigma$ values (right).

## 4 Conclusion

In this paper, we introduced an innovative segmentation framework that leverages an accumulator-based Gaussian map to enhance cell nuclei detection and reduce reliance on manual input. By integrating these Gaussian prompts with multi-scale encoder features through a dynamic cross-attention mechanism, our model effectively focuses on regions likely to contain nuclei, especially in densely packed or overlapping areas where conventional methods often fail.

We evaluated our method on two datasets, showing that it predominantly outperforms state-of-the-art segmentation approaches. The results underscore our method's robustness in handling complex imaging scenarios, demonstrating its practical value for real-world medical applications. Our framework represents a significant advancement in automated medical image segmentation. Future work will target noisy and large-scale datasets to further enhance clinical applicability.

## Acknowledgments and Disclosure of Funding

This work was supported by Institute of Information and communications Technology Planning & Evaluation (IITP) grant funded by the Korea government (MSIT) (No.RS-2022-00155915, Artificial Intelligence Convergence Innovation Human Resources Development (Inha University).

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
