# OpenReview forum: "Centroid-Aware Gaussian Prompt Learning with Erosion-Guided Accumulator for Robust Semantic Cell Segmentation"
_MICCAI.org/2025/Workshop/COMPAYL — COMPAYL 2025_

### Official Review · Reviewer_i1yW · 2025-07-08
**Review of Centroid-Aware Gaussian Prompt Learning with Erosion-Guided Accumulator for Robust Semantic Cell Segmentation**

**Rating:** 5
**Confidence:** 3

**Review:**

Summary:
This manuscript presents a novel and robust cell segmentation method which leverages gaussian prompt learning with an erosion-guided accumulator. The authors demonstrate robust cell segmentation of two challenging datasets. The results presented exceed the performance of existing segmentation models without manual prompting. Further, the authors note that this enhanced model performance is achieved earlier than existing models, demonstrating computational efficiency. The authors conclude by suggesting that this method represents a significant advancement in automated medical image segmentation.

This manuscript is clearly written and easily accessible for the reader. The authors take time to adequately address the current models and literature in the field, providing a comprehensive overview of the current challenges within medical image segmentation. The work herein combines novel concepts to address current challenges within the field and reports an improved segmentation model.

Strengths:
1. The model demonstrates superior segmentation when compared to state-of-the art segmentation models. This is further achieved with fewer parameters and FLOPs, demonstrating advantages in computational efficiency.
2. This model does not require manual prompting, making it less labour intensive and reducing human bias.
3. The model does not rely on the generation of ground truth images, but generates binary masks from the input images, streamlining the workflow.

Weaknesses:
1. Through utilizing gaussian mapping, noisy input images with staining artifacts will result in erroneous segmentation maps. Thus, caution must be applied when handling difficult datasets.
2. Despite its application in two densely packed datasets, the wider application of this segmentation model is not addressed. The manuscript would benefit from additional examples of cell segmentation from diverse tissue types to fully convince the reader of its generalizable utility.
3. The authors note that the model presented is a “significant advancement” in cell segmentation compared to existing models, however the reported improvements of 0.16% and 0.19% remain small and perhaps “significant advancement” may be an overstatement.
4. The cross attention illustration in Figure 1 is not explained in the figure legend.

---

### Official Review · Reviewer_eHwr · 2025-07-20
**Centroid-Aware Gaussian Prompt Learning with Erosion-Guided Accumulator for Robust Semantic Cell Segmentation**

**Rating:** 5
**Confidence:** 4

**Review:**

Summary:
This paper proposes a novel prompt-based approach for semantic cell segmentation by generating centroid-guided Gaussian maps through iterative erosion and accumulation. These maps serve as spatial priors fused with encoder features via cross-attention in a UNet with PVTv2 backbone. The method achieves top results on DSB18 and CoNSeP datasets, surpassing or matching current SOTA models while reducing computational cost.

Strengths:

Innovative use of accumulator-based Gaussian maps to eliminate manual prompting.

Weaknesses:

No analysis of failure modes (e.g., noisy backgrounds, false centroids).

Method is only evaluated on two binary-class datasets; generalization to multi-class segmentation tasks (e.g., tumor microenvironment, stroma vs. tumor vs. immune) is not explored.